# UNCERTAINTY-AWARE SEARCH AND VALUE MODELS

## ABSTRACT

Value model-guided search is effective in steering the generation but suffers from verifier failures: imperfect verifiers may mistakenly prune all the valid paths. This limitation could arise from reliability degradation in value models in unseen reasoning paths. To address this, we propose an uncertainty-aware search framework that includes two key components: (1) uncertainty-aware value models that incorporate uncertainty into predictions, and (2) an uncertainty-aware selection process using the proposed efficient Group Thompson Sampling algorithm. Experiments on two In-Distribution (ID) settings (GSM8K and MATH) and three Out-Of-Distribution (OOD) settings (e.g. AIME25 and Minerva Math) show that our method mitigates verifier failures. This work establishes the first systematic integration of uncertainty quantification in LLM search paradigms.

## 1 INTRODUCTION

Test-time scaling (Brown et al., 2024; Snell et al., 2024; Wu et al., 2024) boosts performance significantly on multi-step mathematical reasoning tasks (Cobbe et al., 2021; Hendrycks et al., 2021). Value Model (VM)-guided search (Yu et al., 2024; Wan et al., 2024) efficiently solves more problems by steering the generation toward more effective reasoning paths.

However, a recent study (Yu et al., 2025) identifies that VM failures could hinder conventional VM-guided search: During the evaluation and selection of candidates in the search process, when VMs underestimate the values of promising candidates and misidentify them, the selection fails, ultimately leading to search failures.

**Uncertainty-Aware Modeling** To mitigate such failures, it is crucial to address unreliable VM predictions at the selection stage. Since VMs rely heavily on training data, they are more likely to provide unreliable predictions when encountering unseen data during inference. This can be addressed by incorporating **uncertainty** to reflect the reliability of VM evaluations, particularly for candidates underrepresented in the training data. Intuitively, candidates less frequently seen during training have higher uncertainty, indicating lower VM reliability and greater risk of selection failures.

To capture uncertainty and develop **Uncertainty-Aware Value Models (UVMs)**, we employ Ensemble++ architecture (Li et al., 2024a;b) to model value distributions, which encapsulate the inherent uncertainty, with a more dispersed distribution indicating higher uncertainty. The trained UVM allows us to evaluate candidates while accounting for uncertainty by sampling from the value distribution.

**Uncertainty-Aware Selection During Search** Leveraging the accessibility to uncertainty-aware value distributions, we develop an efficient algorithm for uncertainty-aware candidate selection. We propose the **Group Thompson Sampling** algorithm, an innovative extension of Thompson Sampling (Thompson, 1933; Russo et al., 2018). This algorithm selects candidates based on their probability of being the optimal choice within the candidate set, i.e. top-1 probability. This method is effective and remains efficient, without the need for explicit top-1 probability estimation. Empirical results demonstrate that our approach outperforms the conventional VM, especially in out-of-distribution (OOD) settings.

In summary, our **contributions** are as follows: (1) We introduce an uncertainty-aware search framework that incorporates uncertainty quantification into the selection stages during the search process,

enhancing the search scaling effectiveness (2) We develop uncertainty-aware value models, equipping VMs with uncertainty quantification (3) We propose the Group Thompson Sampling algorithm, enabling efficient uncertainty-aware candidate selection.

## 2 BACKGROUND

This section first defines the problem and introduces the primary solution framework – search. Then, we introduce outcome value models employed in search, and highlight the issues associated with search and value models.

**Problem definition**   A mathematical reasoning question $q$ requires both the intermediate steps and the final answer as output: The solution path is represented as $S = [s^1, \ldots, s^T, a]$, where $s^i$ is the i-th step, $a$ is the answer, and $T$ is the step count.

**Search**   Search explores correct solutions more efficiently by guiding the process towards more effective paths during the generation. This is achieved through alternating generation and selection stages. In each generation stage, $K$ partial path candidates $\mathbb{S}^{(1:t)} = \left\{ S_k^{(1:t)} \right\}_{k=1}^K$ are produced, where $S_k^{(1:t)} = [s_k^1, \ldots, s_k^t]$ is the $k$-th partial path, and these candidates are then sent to the selection stage. The selection stage then evaluates and selects promising candidates while pruning unpromising ones. The selected candidates are passed to the next generation stage. This process continues until completion. Following Yu et al. (2025), we adopt the step-level beam search framework in this paper, which parallel explores and finally produces $b$ solution paths. See details in Appendix A.3.

**OVM, VM failures, and selection failures**   The Outcome-supervised Value Model (Yu et al., 2024) trains a value model to evaluate each candidate by predicting the probability of it reaching a correct answer. Then, it selects candidates with the highest predicted values from the set $\mathbb{S}^{(1:t)}$. However, the imperfect value model may misidentify and incorrectly rank promising paths, improperly pruning these promising paths, which leads to selection failures (Yu et al., 2025).

## 3 UNCERTAINTY-AWARE VALUE MODELLING

In this section, we first explain the motivation for uncertainty-aware value modelling. Then, we describe the technique used to implement the uncertainty-aware value model in Section 3.1 and how to utilize it in Section 3.2. Finally, we introduce the training process in Section 3.3.

**Motivation**   The performance of VMs heavily depends on the training data. Quantifying uncertainty can reveal the sufficiency of similar training data. Specifically, when sufficient similar data is available in training, the VM offers low-uncertainty, reliable predictions during testing. Conversely, scarce similar data leads to high uncertainty and reduced prediction reliability. By quantifying this uncertainty and evaluating candidates in an uncertainty-aware manner, we can make more informed decisions during the search process.

### 3.1 UNCERTAINTY-AWARE VALUE MODEL

**Ensemble++**   Ensemble++ (Li et al., 2024a;b) is an ensemble-based approach that captures data uncertainty by modelling the posterior distribution. When test data resembles sufficiently seen data, the posterior distribution is concentrated; otherwise, it is more dispersed. This approach is simple to implement, requiring only a learnable linear transformation of the existing representation $\mathbf{x}$. It learns to map a predefined distribution $p_\zeta$, like a Gaussian, to the target posterior distribution.

**UVM**   We borrow Ensemble++ (Li et al., 2024a;b) to model uncertainty-aware values as illustrated in Figure 1 (i). There are three processes involved in UVM: (1) representation encoding (2) index sampling (3) index mapping

**Representation encoding**: The last hidden states from a LLM backbone (parameterized by $\boldsymbol{\theta}$) serve as the representation $\mathbf{x}$. Specifically, for a given question $q$ and a partial path $S^{(1:t)}$, the

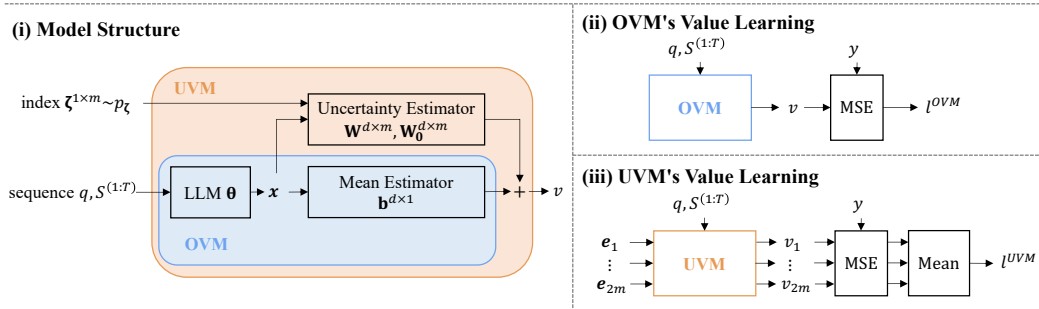

Figure 1: Illustration of the **UVM** structure, value learning process, and its relationship to **OVM**: (i) This figure shows how **UVM** extends **OVM** by adding uncertainty with minimal additional parameters. The blue branch represents **OVM**, which computes a mean value for a sequence. The orange branch introduces the uncertainty term in **UVM**, calculated using parameters $\mathbf{W}$ and $\mathbf{W}_0$. This uncertainty term varies with the input index $\boldsymbol{\zeta}$, leading to diverse posterior value samples. The process of using **UVM** is simple: **UVM** derives a mean value like **OVM**, but also samples from a fixed distribution and adds the uncertainty term. (ii)-(iii) For training, **UVM** uses the same training set as **OVM**, but samples 2m posterior values $[v_1, \ldots, v_{2m}]$ using a discrete coordinate distribution $[\boldsymbol{e}_1, \ldots, \boldsymbol{e}_{2m}]$, rather than estimating a single value. The model is trained by averaging the MSE over these posterior samples.

representation $\mathbf{x}$ is obtained as:

$$\mathbf{x} = \text{LLM}(q, S^{(1:t)}; \boldsymbol{\theta}) \tag{1}$$

**Index sampling**: This process samples an index from the predefined distribution, i.e. $\boldsymbol{\zeta} \sim p_{\boldsymbol{\zeta}}$

**Index mapping**: The index is mapped to the posterior value by summing a mean value term and an uncertainty term: [1]

$$v = \underbrace{\mathbf{x}\mathbf{b}}_{\text{mean estimator}} + \underbrace{\mathbf{x}(\mathbf{W} + \mathbf{W}_0)\boldsymbol{\zeta}^T}_{\text{uncertainty estimator}} \tag{2}$$

Through these processes, a trained UVM maps the predefined distribution $p_{\boldsymbol{\zeta}}$ to the posterior value distribution $p(v|q, S^{(1:t)})$.

**Additional parameters**  Here, $\mathbf{W}^{d \times m}$ and $\mathbf{b}^{d \times 1}$ are learnable parameters, while $\mathbf{W}_0^{d \times m}$ are frozen parameters that are randomly initialized. $d$ represents the dimension of the backbone's hidden states $\mathbf{x}^{1 \times d}$. $m$ is a hyperparameter for the dimension of the index $\boldsymbol{\zeta}^{1 \times m}$. Notably, UVM is simple and straightforward to implement on top of OVM, requiring only an additional linear transformation.

**UVM architectural insight**  It extends conventional approaches through a dual-branch architecture, as shown in Figure 1:

- *Deterministic Branch (blue)*: Maintain standard value estimation, equivalent to OVM
- *Uncertainty Branch (orange)*: Learn distribution through ensemble perturbations

**Intuitive explanation**  UVM can be regarded as a last-layer ensemble of $m$ components, controlled by the index vector $\boldsymbol{\zeta}^{1 \times m}$: (1) When $\boldsymbol{\zeta}^{1 \times m}$ is a zero vector $[0, \ldots, 0]$, UVM retains only the deterministic branch, degenerating to OVM (2) When $\boldsymbol{\zeta}^{1 \times m}$ is a one-shot index vector, with a 1 in the $i$-th position and 0s elsewhere, it queries the $i$-th component for the prediction (3) When $\boldsymbol{\zeta}^{1 \times m}$ is a non-one-shot vector, it is equivalent to use a linear combination of the $m$ components.

---

[1]For simplicity, we present the main expression and omit the hyperparameters involved in the practical implementation. Details on the practical implementation can be found in A.2.

## 3.2 Accessing Values

Although the explicit formalization of $p(v|q, S^{(1:t)})$ is unavailable, we can sample values and compute the distribution's mean and standard deviation.

**Sampling from posterior value distribution**   To derive a posterior value, we (1) sample an index $\zeta \sim p_\zeta$ and (2) map it to the value sample $v$ using Equation 1-2. Notably, deriving multiple posterior values only requires one-time representation encoding, i.e. forward pass through the LLM backbone (Equation 1), followed by multiple posterior value mappings (Equation 2) with repeated index sampling and mapping.

**Capturing the distribution's mean and standard deviation**   The mean is obtained by using a zero vector as the index $\zeta$ and mapping it. The standard deviation is estimated by sampling multiple posterior values, which involves repeated index sampling and mapping.

## 3.3 Uncertainty-Aware Value Learning

UVM does not require any additional training data and can use the same training dataset as OVM. The training dataset construction is described in Appendix A.1, which consists of $(q, S^{(1:T)}, y)$ tuples, where $y$ denotes the correctness of the final answer.

Given the same training dataset, the key difference in value learning between OVM and UVM lies in their training objectives:

**Training loss of OVM**   OVM learns single-point estimates (Figure 1 (ii)). Its loss is the Mean Squared Error (MSE) with respect to the binary label $y$, for each $(q, S^{(1:T)}, y)$ tuple:

$$L^{\text{OVM}}(q, S^{(1:T)}, y) = \sum_{t=1}^{T} (\text{OVM}(q, S^{(1:t)}) - y)^2 \tag{3}$$

where $\text{OVM}(\cdot)$ evaluates and maps a given partial path $S^{(1:T)}$ and the question $q$ to a value scalar.

**Training loss of UVM**   UVM is learning a posterior value distribution, which complicates its learning (Figure 1 (iii)). Following Ensemble++ (Li et al., 2024a;b), a discrete coordinate distribution is used in training. This distribution consists of $2m$ one-hot index vectors $[\boldsymbol{e}_1, \ldots, \boldsymbol{e}_m, \boldsymbol{e}_{m+1}, \ldots, \boldsymbol{e}_{2m}]$. For each $i = 1, \ldots, m$, the $i$-position of $\boldsymbol{e}_i$ is 1, while of $\boldsymbol{e}_{m+i}$ is -1, with 0s elsewhere. The training loss for each $(q, S^{(1:T)}, y)$ tuple is

$$L^{\text{UVM}}(q, S^{(1:T)}, y) = \sum_{t=1}^{T} \frac{1}{2m} \sum_{i=1}^{2m} (\text{UVM}(q, S^{(1:t)}, \boldsymbol{e}_i) - y)^2 \tag{4}$$

Notably, the gradients of the term $\mathbf{W}$ are stopped to prevent them from propagating to the backbone parameters $\boldsymbol{\theta}$.

**Explanation on the learning process**   Under this objective, the posterior uncertainty term $\mathbf{W}$ learns to offset the random prior $\mathbf{W}_0$, while the parameters $\mathbf{b}$ focus on modeling the mean of the posterior distribution. When posterior learning is sufficient, the posterior uncertainty term $\mathbf{W}$ reduces the noise introduced by the prior $\mathbf{W}_0$, leading to a low-variance posterior distribution. If learning is insufficient, the prior $\mathbf{W}_0$ dominates, causing a high-variance posterior distribution.

## 4 Uncertainty-Aware Selection

This section describes (1) the implementation of posterior value sampling during inference and (2) uncertainty-aware selection with the accessibility to the posterior value distribution. We propose a novel algorithm, Group Thompson Sampling, to effectively and efficiently select multiple candidates.

**Posterior value sampling of UVM during inference**    Following Ensemble++ (Li et al., 2024a;b), a $m$-dimensional continuous Gaussian distribution is used as the index distribution $p_\zeta$ to access the expressive posterior value distribution during inference, as described in Section 3.2. Intuitively, we are training $m$ individual components during value learning, and then combine them linearly to make predictions during inference.

**Top-1 probability for candidate selection**    For each step $t$, we evaluate and rank $K$ candidates in the set $\mathbb{S}^{(1:t)}$ based on their posterior value distribution. Specifically, we assess the probability of each candidate being the best, i.e. the likelihood of it having the highest value, within the set, referred to as the top-1 probability

$$p(S_i^{(1:t)} \text{ is the best among } \mathbb{S}^{(1:t)})$$

$$= \mathbb{E}_{v_i \sim p(v|q,S_i^{(1:t)}), v_j \sim p(v|q,S_j^{(1:t)}) \ \forall j \neq i} \left[ \prod_{\substack{j \neq i}} \mathbb{I}(v_i \geq v_j) \right] \tag{5}$$

We perform candidate selection by sampling from the top-1 probability distribution

$$S^{(1:t)} \sim p(S^{(1:t)} \text{ is the best among } \mathbb{S}^{(1:t)}) \tag{6}$$

We use top-1 probability for candidate selection instead of the well-known Upper Confidence Bound (UCB) [2] for two main reasons:

- *Top-1 probability better captures the distribution's characteristics, balancing mean and variance*. In candidate selection, it is important to balance the predicted mean value with the uncertainty (i.e., the distribution's variance). Candidates with extreme uncertainty will be prioritized by UCB, potentially overlooking the mean value prediction in such cases. In contrast, top-1 probability provides a more balanced approach that not overemphasizes uncertainty.

- *Top-1 probability inherently ranks candidates against each other*. UCB evaluates each candidate independently, without considering its relation to others. In contrast, top-1 probability intrinsically compares each candidate to all others in the set, ranking them based on the likelihood of being the best in the set.

However, explicitly estimating the top-1 probability for each candidate requires multiple posterior value samples, which involves repeated index sampling and mapping. This process can be computationally expensive.

**Group Thompson Sampling: An efficient algorithm for group selection based on top-1 probability**    Group Thompson Sampling is an extension of the Thompson sampling algorithm, designed to efficiently select a group of candidates based on their top-1 probability. Thompson sampling (Thompson, 1933; Russo et al., 2018) is a straightforward method for selecting a single candidate, by performing one posterior sampling and choosing the one with the highest sampled value

$$v_i \sim p(v|q, S_i^{(1:t)}), \forall i$$
$$S_i^{(1:t)} = \operatorname*{argmax}_i v_i \tag{7}$$

To extend this for selecting multiple candidates, we introduce Group Thompson Sampling algorithm in Algorithm 1. This method repeats Thompson sampling $b$ times to select $b$ candidates, incorporating a mechanism to avoid duplication.

**Algorithmic innovation**    Group Thompson Sampling addresses two key challenges in uncertainty-aware search:

- *Computational efficiency*: Avoid explicit top-1 probability estimation through smart sampling

- *Comprehensive selection range*: Ensure that candidates with varying levels of uncertainty are appropriately considered for selection through an uncertainty-aware stochastic selection mechanism

---

[2]UCB scores each candidate using the sum of an exploitation term and an exploration term, i.e. the sum of mean values and standard variation in our case.

---

**Algorithm 1 Group Thompson Sampling**

---

**Input:** Question $q$, candidates $\{S_1, \ldots, S_K\}$, beam size $b$
**Output:** $b$ selected candidates
**Model:** UVM
**Hyperparameter:** Maximum tries $T^{max}$
1: Sample index $\xi \sim p_\xi$ and select $i = \underset{i=1,\ldots,K}{\operatorname{argmax}} \ \mathrm{UVM}(S_i; q, \xi)$
2: Initialize selected set $\mathcal{I} \leftarrow \{i\}$
3: **repeat**
4: $\quad$ Sample $\xi \sim p_\xi$ and select $i = \underset{i=1,\ldots,K}{\operatorname{argmax}} \ \mathrm{UVM}(S_i; q, \xi)$
5: $\quad$ Add $i$ to $\mathcal{I}$ if $i \notin \mathcal{I}$; otherwise try 4 again
6: $\quad$ After $T^{max}$ tries, instead sample non-repeated $i$ uniformly
7: **until** there are $b$ selected candidates
$\quad$ **return** $\{S_i | i \in \mathcal{I}\}$

---

## 5 EXPERIMENT RESULTS

This section outlines our experiment settings and presents the results of overall performance.

### 5.1 EXPERIMENTAL SETTINGS

**Benchmarks** We compare our method, UVM-guided search, with the conventional OVM-guided search, which does not incorporate uncertainty. We consider both ID and OOD scenarios, including traditional OOD and Randomized Token Noise OOD (RTN-OOD) settings.

- *ID settings*: We run experiments on GSM8K (Cobbe et al., 2021) and MATH (Hendrycks et al., 2021). For training, we use the official training split of each dataset; for evaluation, we use the official GSM8K test set and MATH500 (Lightman et al., 2024).

- *Traditional OOD settings*: We run experiments on two additional benchmarks: AIME25[3] and MinervaMath[4].

- *RTN-OOD*: We define an RTN-OOD setting in which the VM training distribution consists of sequences whose tokens are sampled from the tokenizer vocabulary. Evaluation is conducted on MATH500, referred to as RTN-MATH. This enables a controlled comparison to ID MATH, focused on extreme training set shift, by varying only the training distribution.

**Models** On GSM8K, we use Mistral 7B (Jiang et al., 2023) and Qwen2.5-Math 7B (Yang et al., 2024); on MATH, we use Qwen2.5-Math 7B and Qwen3 8B (Yang et al., 2025). For all OOD settings, we use Qwen2.5-Math 7B. [5]

**Evaluation** We evaluate our method using two metrics: Coverage and precision. Coverage measures the fraction of problems for which the correct solution is covered by the generated paths, i.e., at least one sampled path is correct. This is also known as pass@k, where $k$ denotes the number of generated paths. [6] For simplicity, precision is quantified under the majority voting strategy (Wang et al., 2023) as the fraction of problems for which the majority answer among the generated paths is correct.

We conduct the main experiments under the largest inference setting in Yu et al. (2025), using a beam size of 32 and a candidate size of 256. Each experiment is repeated three times, and we report the average metrics along with their standard deviation.

---

[3] https://huggingface.co/datasets/math-ai/aime25

[4] https://huggingface.co/datasets/math-ai/minervamath

[5] Qwen3-8B is adjusted to produce traditional Chain-of-Thought (CoT) in this paper; observing no gains over Qwen2.5-Math, we therefore use Qwen2.5-Math for all OOD settings.

[6] In our paper, $K$ represents the candidate size during the intermediate search process, rather than the number of final produced paths. To avoid confusion, we use the term "coverage".

## 5.2 IMPLEMENTATION

**Training generators**   We train the three base models (i.e. Mistral 7B, Qwen2.5-Math 7B, and Qwen3 8B) on the official training set of GSM8K and MATH, respectively. We use the newline character as the marker for the end of each step. Supervised fine-tuning is performed for 2 epochs with a batch size of 128. We use a linear learning rate scheduler with a maximum learning rate of 2e-6 for Mistral 7B, 2e-5 for Qwen2.5-Math 7B, and 1e-5 for Qwen3 8B. The AdamW optimizer (Loshchilov & Hutter, 2019) is used for training.

**Building training dataset for UVMs**   The dataset construction process is introduced in Appendix A.1. We sample 50 solution paths per problem in the training set. For dataset collection, we use a decoding temperature of 0.7, top-k set to 50, and the maximum new token length set to 400 on GSM8K; we use a temperature of 1, top-p set to 0.98, and the maximum new token length set to 2000 on MATH. We apply vllm (Kwon et al., 2023) to accelerate the generation process.

**Training UVMs/OVMs**   To construct UVMs, we set the number of components to 10. UVMs are initialized from the corresponding generator checkpoints and trained for one epoch, using the same backbone learning rate scheduler. The maximum learning rate for the uncertainty-aware value head is set to 2e-3 for Mistral 7B, 2e-4 for Qwen2.5-Math 7B, and 1e-3 for Qwen3-8B. The batch size is set to 128 on GSM8K and to 512 on MATH. The optimizer used for training is AdamW. After training, we derive OVMs by setting $\zeta = 0$.

**Step-level beam search**   The decoding hyperparameters mirror those used for UVM training data collection; we set the maximum number of steps to 10 on GSM8K and 20 on MATH, and for AIME25 and Minerva Math, we adopt the MATH settings with two exceptions: a 4096 maximum new token length and a 50 maximum number of steps.

## 5.3 OVERALL PERFORMANCE

We present the overall performance in Table 1 and Table 2.

**Coverage gains with UVM**   *UVM-guided search consistently outperforms conventional OVM-guided search on coverage in both ID and OOD settings*. In ID settings, the average coverage across models and benchmarks increases by 2.2%. For example, it raises Mistral's coverage on GSM8K by about 4.3% and Qwen2.5-Math's coverage on MATH by about 1.3%. Gains are larger in the OOD settings: average coverage rises from 29.4% to 41.8%. For instance, coverage on Minerva Math improves by roughly 20%. This stronger OOD performance aligns with UVM's design objective—capturing uncertainty to better handle unseen data.

**Precision outcomes with UVM under majority voting**   *UVM-guided beam search does not guarantee higher precision under majority voting*. In the ID settings, the average precision across models and benchmarks decreases by 1.3%, whereas in the OOD settings it increases by 3.5%. Although UVM improves coverage, these gains translate less directly to precision, leaving a substantial gap. This is consistent with intuition: because UVM explicitly models uncertainty, it naturally explores a wider range of solution paths than OVM, which can reduce the likelihood that correct paths form a majority under voting. Even so, we observe positive gains in the OOD setting, demonstrating the effectiveness of our method on unseen data.

Table 1: Performance of UVM-guided search relative to OVM-guided baseline in the ID settings ('mst': 'mistral', 'qwm': 'qwen2.5-math', 'qw3': 'qwen3')

|  |  | GSM8K | | MATH | | Average |
|  |  | mst | qwm | qwm | qw3 | |
|---|---|---|---|---|---|---|
| Coverage | OVM | 87.3% ± 0.4% | 95.2% ± 0.2% | 80.1% ± 0.4% | 79.4% ± 0.9% | 85.5% |
|  | UVM | 91.6% ± 0.4% | 96.9% ± 0.1% | 81.4% ± 0.7% | 80.5% ± 0.4% | 87.6% |
| Precision | OVM | 82.2% ± 0.6% | 93.6% ± 0.2% | 66.5% ± 0.7% | 70.9% ± 0.7% | 78.3% |
|  | UVM | 82.8% ± 0.3% | 92.5% ± 0.3% | 66.7% ± 0.8% | 68.5% ± 0.8% | 77.6% |

Table 2: Performance of UVM-guided search relative to OVM-guided baseline in the OOD settings (Qwen2.5-Math 7B)

|  |  | AIME25 | Minerva Math | RTN-MATH | Average |
|---|---|---|---|---|---|
| Coverage | OVM | 8.9% ± 1.6% | 23.8% ± 1.8% | 55.6% ± 0.7% | 29.4% |
|  | UVM | 8.9% ± 1.6% | 43.6% ± 1.8% | 72.8% ± 0.8% | 41.8% |
| Precision | OVM | 5.6% ± 1.6% | 13.2% ± 1.2% | 42.1% ± 0.2% | 20.3% |
|  | UVM | 2.2% ± 3.2% | 16.5% ± 2.2% | 52.7% ± 1.0% | 23.8% |

These results highlight the effectiveness of our method in addressing verifier failures, especially in the OOD scenarios.

## 6  ANALYSIS

This section presents ablation studies on the design choices of uncertainty-aware selection and studies the effectiveness of UVM across various inference settings.

### 6.1  MORE INFERENCE SETTINGS

In this section, we analyze the effectiveness of our method under more inference settings in Table 3. Similar to Yu et al. (2025), we fix the number of generated paths per beam $K/b$ at 8 and conduct experiments with beam sizes of 1, 2, and 16. Under this setup, we obtain two small configurations (1 and 2) and two large configurations (16 and 32).

Table 3: Increased average coverage of UVM-guided search over OVM-guided search ('mst': 'mistral', 'qwm': 'qwen2.5-math', 'qw3': 'qwen3')

| Beam Size | GSM8K | | MATH | | RTN-MATH |
|---|---|---|---|---|---|
|  | mst | qwm | qwm | qw3 | qwm |
| 1 | -7.6% | +0.4% | -0.6% | -10.3% | -0.2% |
| 2 | -2.2% | +0% | -2.2% | -6.2% | +1.1% |
| 16 | +4.7% | +0.6% | +0% | +0.8% | +15.5% |
| 32 | +4.3% | +1.7% | +1.3& | +1.1% | +17.2% |

**Effectiveness across configurations** *UVM-guided search favors larger configurations*. As shown in Table 3, UVM-guided search outperforms OVM-guided search at beam sizes of 16 and 32–for example, it yields a 4.7% gain on GSM8K using Mistral and a 15.5% gain on RTN-MATH using Qwen2.5-Math, with a beam size of 16. In contrast, the gains are limited or even negative at beam sizes of 1 and 2; for instance, on GSM8K with Mistral the gains are -7.6% and -2.2%, respectively. This pattern aligns with intuition: UVM explicitly models uncertainty and sometimes promotes high-uncertainty candidates; while these can contain correct reasoning paths, this is not guaranteed. Consequently, larger configurations provide the necessary room for UVM's uncertainty-aware selection to realize its advantages.

**Effectiveness under distribution shift** *UVM contributes more under distribution shift*. As shown in Table 3, the gains rise from 0% and 1.3% with an ID training set to 15.5% and 17.2% at beam sizes of 16 and 32 when the model is trained on the RTN dataset. This highlights the effectiveness of UVM in handling uncertainty when the training set is a severe mismatch with the test set.

### 6.2  ABLATION STUDY

In this section, we conduct ablation studies on the methods of uncertainty-aware selection and demonstrate the superiority of the Group Thompson Sampling algorithm in Table 4.

**Baselines**  We consider two deterministic baselines for uncertainty-aware selection: UCB ranking and the naive top-1 probability ranking.

- *UCB ranking*: UCB scores each candidate by summing the mean and standard deviation of the posterior value distribution, which are computed using 100,000 sampled posterior values. Then, candidates with the highest UCB scores are selected.
- *Naive top-1 probability ranking*: This method explicitly calculates the probability of each candidate being the best, as described in Equation 5, using 100,000 sampled posterior values. It ranks and selects candidates based on the highest probabilities, rather than sampling from the probability distribution.

**Top-1 probability ranking v.s. UCB ranking**  *Top-1 probability is better for candidate evaluation*. As shown in Table 4, the top-1 probability ranking surpasses the UCB, demonstrating the effectiveness of using top-1 probability for candidate evaluation.

**Group Thompson Sampling v.s. top-1 probability ranking**  *Group Thompson Sampling is an effective algorithm for uncertainty-aware selection*. As illustrated in Table 4, our proposed Group Thompson Sampling algorithm achieves higher coverage than the naive top-1 probability ranking. Furthermore, by sampling candidates according to the underlying top-1 probability distribution, Group Thompson Sampling eliminates the need for explicit top-1 probability estimation, making it more efficient.

These results show the superiority of our Group Thompson Sampling algorithm: It achieves higher coverage than the other baselines while requiring fewer computations, enhancing both effectiveness and efficiency compared to other uncertainty-aware selection strategies.

Table 4: Ablation on design choices of uncertainty-aware selection under coverage

|  | GSM8K | | MATH | | Average |
| --- | --- | --- | --- | --- | --- |
|  | mst | qwm | qwm | qw3 |  |
| UCB Ranking | 86.3% ± 0.4% | 95.7% ± 0.3% | 81.7% ± 0.7% | 79.7% ± 0.3% | 85.9% |
| Top-1 Probability Ranking | 88.6% ± 0.5% | 96.2% ± 0.2% | 81.7% ± 0.9% | 79.5% ± 1.2% | 86.5% |
| Group Thompson Sampling | 91.6% ± 0.4% | 96.4% ± 0.2% | 81.4% ± 0.7% | 80.5% ± 0.4% | 87.5% |

## 7 DISCUSSION

Quantifying uncertainty in search-based methods has several practical implications for LLM reasoning tasks as follows: (1) *Discovering correct solutions*: Correct solutions may lie along paths that are underrepresented in the training data. By incorporating uncertainty, our approach increases the likelihood of discovering these solutions. (2) *Improved performance with limited additional resources*: Uncertainty-aware search methods enhance performance without a significant increase in computational resources. This is because the process of quantifying uncertainty is computationally inexpensive. (3) *Adaptability to real-world applications*: Real-world applications often encounter OOD data. Methods that account for uncertainty are better equipped to handle such cases, enabling more reliable performance in deployment scenarios where the data may differ from training distributions.

## 8 CONCLUSION

VM-guided search suffers from search scaling flaws due to the use of imperfect VMs, which could produce unreliable predictions when the evaluated data is underrepresented in the training data. To address these issues, we propose an uncertainty-aware search framework, including training uncertainty-aware VMs and applying uncertainty-aware selection during the search. Experiment results in GSM8K show the effectiveness of our methods.

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

## A APPENDIX

Table 5: Summary of Notations Used in the Paper

| Notation | Description |
| --- | --- |
| $q$ | Mathematical reasoning question requiring a sequence of steps |
| $S$ | Solution path for a question, $S = [s^1, \ldots, s^T, a]$ |
| $s^i$ | $i$-th step in a solution path |
| $a$ | Final answer in a solution path |
| $T$ | Number of steps in a solution path |
| $y$ | Binary label (0 or 1) indicating the correctness of $a$ |
| $S^{(1:t)}$ | Partial solution path up to step $t$, $S^{(1:t)} = [s^1, \ldots, s^t]$ |
| $\mathbb{S}^{(1:t)}$ | Set of candidate partial paths $\mathbb{S}^{(1:t)} = \{S_k^{(1:t)}\}_{k=1}^K$ |
| $K$ | Candidate size |
| $b$ | Beam size |
| $v$ | Value (scalar) of a partial path |
| $p(v|q, S^{(1:t)})$ | Posterior distribution of values for a partial path |
| $m$ | Number of components in UVM head |
| $\mathbf{W}, \mathbf{b}$ | Learnable parameters of UVM head |
| $\mathbf{W}_0$ | Fixed parameters of UVM head |
| $\boldsymbol{\theta}$ | Parameters of the value model backbone |
| $d$ | Dimension of value model backbone's hidden states |
| $\mathbf{x}$ | Last hidden states output by VM's backbone |
| $\boldsymbol{\zeta}$ | Index vector of dimension $m$ |
| $p_{\boldsymbol{\zeta}}$ | Index distribution |

### A.1 CONSTRUCTION OF VMS' TRAINING DATASET

The training dataset is created using the generator and the question-answer pairs. For each pair $(q, a) \in \mathcal{Q}$, the generator produces $n$ solution paths, resulting in a total of $|\mathcal{Q}| \times n$ question-solution

pairs. The label $y$ for each solution $S$ is assigned based on the correctness of the final answer, which is determined by comparing it to the ground truth answer $a$. A label of 1 indicates the answer is correct, while 0 indicates it is incorrect. This process forms a training dataset consisting of $(q, S, y)$ tuples for value model training.

## A.2 DETAILED UVM STRUCTURE

Given the representation $\mathbf{x}$, the sampled index $\boldsymbol{\zeta}$ is mapped to the posterior value sample as:

$$v = \underbrace{\mathbf{xb}}_{\text{mean estimator}} + \underbrace{\mathbf{x}(u\mathbf{W} + p_0\mathbf{W}_0)\boldsymbol{\zeta}^T}_{\text{uncertainty estimator}} \tag{8}$$

Here, $u$ and $p_0$ are hyperparameters. Specifically, $u$ controls the tradeoff between the uncertainty terms $\mathbf{W}, \mathbf{W}_0$ and the mean value term $\mathbf{b}$, and $p_0$ controls the tradeoff between the posterior term $\mathbf{W}$ and the prior term $\mathbf{W}_0$.

## A.3 STEP-LEVEL BEAM SEARCH

The algorithm is shown in Algorithm 2.

---

**Algorithm 2** Step-Level Beam Search

---

**Input:** Question $q$, Beam size $b$, Candidate size $K$, Maximum step count $T^{max}$
**Output:** $b$ solution sequences for $q$
**Model:** Generator and VM
1: Initialize sequences $\mathbb{S} \leftarrow \{\}$
2: Sample the first steps $\{s_1^1, \ldots, s_K^1\}$
3: Select $b$ steps via SELECTION($q$, $\{s_1^1, \ldots, s_K^1\}$, $b$, VM) and add to $\mathbb{S}$
4: $t \leftarrow 1$
5: **while** any sequence of $\mathbb{S}$ is not complete and $t < T^{max}$ **do**
6:     $\mathbb{S}_{\text{next}} \leftarrow \{\}$
7:     **for** $S^{(1:t)}$ in $\mathbb{S}$ **do**
8:         **for** $i = 1$ to $K/b$ **do**
9:             $S_i^{(1:t+1)} = \text{Generator}(S_i^{(1:t)}; q)$
10:             $\mathbb{S}_{\text{next}} \leftarrow \mathbb{S}_{\text{next}} + S_i^{(1:t+1)}$
11:     $\mathbb{S} \leftarrow \text{SELECTION}(q, \mathbb{S}_{\text{next}}, b, \text{VM})$
12:     $t \leftarrow t + 1$
    **return** $\mathbb{S}$

---

## A.4 LLM USAGE

We use GPT5 to aid and polish writing.

