# OpenReview forum: "Uncertainty-Aware Search and Value Models in LLMs"
_ICLR.cc/2026/Conference — ICLR 2026 Conference Withdrawn Submission_

### Official Review · Reviewer_dD8N · 2025-10-29

**Soundness:** 3
**Presentation:** 3
**Contribution:** 2
**Rating:** 4
**Confidence:** 3

**Summary:**

This paper aims to solve the reliability problem of standard value models (OVM) in the guided search process, which arises from insufficient training data coverage. The authors propose an "uncertainty-aware" search framework, the core of which is an Uncertainty Value Model (UVM) using the Ensemble++ architecture to quantify prediction uncertainty. It combines the Group Thompson Sampling algorithm to balance value and uncertainty, thereby selecting more promising reasoning paths. This method aims to reduce the risk of pruning correct solutions due to incorrect evaluations by the value model.

**Strengths:**

1.  Clear and Important Problem Definition: The paper clearly identifies a core flaw in current value model-guided search—"verifier failures," where the value model may incorrectly prune valid solutions due to not having seen certain reasoning paths. This is a very practical and important problem in enhancing the reasoning capabilities of large language models (LLMs).
2.  Methodological Innovation: The paper is the first to integrate uncertainty quantification into the LLM search paradigm. By borrowing from and extending the Ensemble++ architecture, the model can learn a posterior distribution instead of a single value score, thus quantifying the unreliability of evaluating unseen paths.

**Weaknesses:**

1. Limited Novelty: The method primarily combines existing Ensemble++ techniques with OVM. Its core highlight lies more with Ensemble++ itself rather than the UVM framework, thus it is an incremental innovation.
2. Core Assumption is Questionable: The experimental results (Tables 1 and 2) show that the improvement in coverage did not lead to a significant increase in accuracy, and even caused a decrease in accuracy on in-distribution datasets. This diminishes the practical significance of the paper's core motivation to "improve performance by increasing coverage".
3. Insufficient Experimentation: The paper claims that UVM is suitable for larger beam sizes, but the experiments only went up to 32. To fully validate its effectiveness, experiments should be conducted on larger beam sizes (e.g., 64 or 128). The current experiments only show that UVM is not suitable for small beam sizes.
4. Lack of Parameter Analysis: The paper does not provide an analysis of key hyperparameters, such as the length of the index vector m and other trade-off parameters. This raises doubts about the reproducibility and robustness of the results.
5. Layout and Visualization: The overall layout quality of the paper needs improvement, and it lacks necessary visualizations to intuitively demonstrate the significance and effect of its method's components.

**Questions:**

1. The experimental results show that coverage and accuracy do not have a simple positive correlation. Could you further analyze the complex relationship between these two, or consider additional model designs to ensure that improved coverage translates more effectively into improved accuracy?
2. The paper mentions that the UVM framework is suitable for larger beam sizes, but related experiments are missing. Why were tests not conducted on larger beam sizes (e.g., 64 or higher) to support this claim?
3. The model introduces new hyperparameters (like the index vector length m). Can you provide a sensitivity analysis for these hyperparameters to help readers understand their impact on model performance?

---

### Official Review · Reviewer_BZbU · 2025-10-31

**Soundness:** 2
**Presentation:** 2
**Contribution:** 2
**Rating:** 2
**Confidence:** 3

**Summary:**

The paper proposes an uncertainty-aware search framework to solve that imperfect verifiers may mistakenly prune all the valid paths for value model-guided search. The framework includes uncertainty-aware value models and an uncertainty-aware selection process. Experiments on two In-Distribution settings (GSM8K and MATH) and three Out-Of-Distribution settings show that the proposed method mitigates verifier failures.

**Strengths:**

1. The method learns Uncertainty-Aware Value Models to mitigate VM unreliability during inference.
2. With the Group Thompson Sampling, the method leverages the Uncertainty-Aware Value Model to Uncertainty-Aware Selection During Search.

**Weaknesses:**

1. The paper was not written clearly enough.
2. The contribution and motivation of the paper need to be clarified.
3. The Group Thompson Sampling algorithm requires repeated sampling and maximum value operations b times, but the paper lacks analysis and discussion of its cost.
4. The experimental comparative methods are insufficient.

**Questions:**

1. What is the predefined distribution in the Index sampling? Where is it from?
2. The paper does not strongly demonstrate the superiority of the proposed Group Thompson Sampling algorithm. Can more comprehensive evidence be provided?
3. In Section 5, why are Mistral and Qwen3 evaluated on different datasets, respectively, rather than both being tested on the same datasets for comparison?
4. There is a lack of sufficient baseline method comparisons.

---

### Official Review · Reviewer_uwTZ · 2025-10-31

**Soundness:** 2
**Presentation:** 1
**Contribution:** 1
**Rating:** 2
**Confidence:** 3

**Summary:**

Addressing the "verifier failure" problem in value model-guided search, this paper proposes an uncertainty-aware search framework. This framework comprises two key techniques: an uncertainty-aware value model (UVM) and uncertainty-aware selection. Specifically, the UVM employs the Ensemble++ architecture, extending the traditional single-point estimation value model (OVM) into a model capable of modeling the posterior distribution of values, thereby quantifying prediction uncertainty. The uncertainty-aware selection component introduces the Group Thompson Sampling method, which samples from the posterior value distribution and selects the Top-1 Probability path among candidate paths. The paper validates that the proposed method mitigates verifier failures in both In-Distribution and Out-Of-Distribution settings.

**Strengths:**

1. The motivation of this paper is intuitive and valuable.

**Weaknesses:**

1. The discussion of related work is insufficient.
2. The writing of the paper requires further improvement. Abbreviations should be spelled out upon first use (e.g., OVM), and some section arrangements are less reasonable (e.g., Chapter 7 and Chapter 8 could be merged).
3. The proposed method framework requires a generator to produce training data and obtains a discrete coordinate distribution based on the generated data, which incurs significant computational resources and time costs and is unfavorable for the generalization of the method.

**Questions:**

1. If the output x from the LLM is erroneous, will the learned posterior value distribution also be affected?
2. When the proposed UVM encounters tasks similar to those in the dataset, can it distinguish the degree of similarity or importance?
3. In Section 3.1, why was Ensemble++ chosen to capture data uncertainty? The corresponding rationale should be provided.

---

### Official Review · Reviewer_pn8D · 2025-11-01

**Soundness:** 3
**Presentation:** 1
**Contribution:** 1
**Rating:** 0
**Confidence:** 4

**Summary:**

The paper proposes uncertainty aware value model guided search. The method is based on what they call Group Thompson Sampling.

**Strengths:**

Uncertainty awareness is crucial for inference in models where uncertainty varies in wide margins depending on the input.

**Weaknesses:**

The authors are not familiar with the literature. Their innovative sampling algorithm is called Multi-Play Thompson sampling and is well known in the literature: https://arxiv.org/abs/1506.00779 . There are other problems, but this one is sufficient for rejection.

**Questions:**

There are many distributions for which the standard deviation does not quantify the uncertainty, at least not enough to guide search. What makes you conclude that standard deviation is the right quantification in your case?

---

> ### Author Response · Authors · 2025-12-02
>
> 1. **Response to Weakness**
>
> The key distinction lies between a one-shot selection process (https://arxiv.org/pdf/1506.00779) and an iterative selection process (our proposed Group Thompson Sampling).
>
> The selection process in https://arxiv.org/pdf/1506.00779 can be regarded as a top-k Thompson Sampling scheme: it samples one value from the posterior distribution for each candidate and then selects the top K candidates with the highest sampled values from this single draw.
>
> In contrast, our Group Thompson Sampling employs an iterative selection strategy. In each iteration, we sample one value for every candidate and select the candidate with the highest sampled value. This process repeats until K unique candidates have been selected, one per iteration.
>
> 2. **Response to Q1**
>
> In our selection algorithm, Group Thompson Sampling, we do not assume any specific underlying distribution. The uncertainty is quantified using the standard deviation only for the UCB-based selection, which is used as a comparative baseline.

---

### Note · Authors · 2025-12-02

**Comment:**

We would like to sincerely thank the reviewers and editors for their time, efforts, and constructive comments on our manuscript. Their valuable suggestions have provided us with important insights to further improve our work.

Regarding the computational aspect, our proposed structure does not require significant computational resources, as it only involves repeated sampling from a low-dimensional Gaussian distribution followed by mapping through a simple linear structure. Following the reviewers’ suggestions, we plan to conduct an in-depth analysis in this regard.

We will take this opportunity to carefully revise the manuscript, improve the overall writing quality, and include additional analyses to strengthen and clarify our results.

**Withdrawal Confirmation:**

I have read and agree with the venue's withdrawal policy on behalf of myself and my co-authors.